# Unstructured spare time as an international predictor of adolescent crime

David Buil-Gil[1]*, Christopher Birkbeck[2], Dirk Enzmann[3],
Karin Arbach[4], Marina Rezende Bazon[5], Muhamed Budimlić[6], Danielle T. Cooper[7],
Marta Dąbrowska[8], Esther Fernández-Molina[9], Carolyn Gentle-Genitty[10],
Aurea E. Grijalva-Eternod[11], Ewa M. Guzik-Makaruk[12], Sandrine Haymoz[13], Neal Hazel[2],
Markus Kaakinen[14], Janne Kivivuori[14], Iza Kokoravec Povh[15],
Camilla Løvschall Langeland[16], Patrik Manzoni[17], Anna Markina[18], Gorazd Meško[15],
Sandra Kobajica Mišanović[6], Kim Moeller[19], Solbey Morillo-Puente[20], Aušra Pocienė[21],
Zuzana Podaná[22], Juan Antonio Rodríguez[23], Margret Valdimarsdottir[24], Lars Westfelt[25],
Ineke Haen Marshall[26]

1 Department of Criminology, University of Manchester, Manchester, United Kingdom, 2 School of Health and Society, University of Salford, Salford, United Kingdom, 3 Department of Social Sciences, University of Hamburg, Hamburg, Germany, 4 Institute of Psychological Research, National Scientific and Technical Research Council, Córdoba, Argentina, 5 Department of Psychology, University of São Paulo, Monte Alegre, Brazil, 6 Faculty of Criminal Justice, Criminology and Security Studies, University of Sarajevo, Sarajevo, Bosnia and Herzegovina, 7 Criminal Justice Department, University of New Haven, West Haven, Connecticut, United States of America, 8 Department of Criminal Law and Criminology, University of Białystok, Białystok, Poland, 9 Research Group in Criminology and Juvenile Delinquency, University of Castilla-La Mancha, Ciudad Real, Spain, 10 Founder's College, Butler University, Indianapolis, Indiana, United States of America, 11 Centre for Social Sciences and Humanities, University of Guadalajara, Guadalajara, Mexico, 12 Institute of Law, Police Academy in Szczytno, Szczytno, Poland, 13 School of Social Work, University of Applied Sciences and Arts (HES-SO), Fribourg, Switzerland, 14 Institute of Criminology and Legal Policy, University of Helsinki, Helsinki, Finland, 15 Faculty of Criminal Justice and Security, University of Maribor, Ljubljana, Slovenia, 16 Department of Criminology and Sociology of Law, University of Oslo, Oslo, Norway, 17 School of Social Work, ZHAW Zurich University of Applied Sciences, Zürich, Switzerland, 18 School of Law, University of Tartu, Tallinn, Estonia, 19 Department of Criminology, Malmö University, Malmö, Sweden, 20 International Bridge University, Doral, Florida, United States of America, 21 Institute of Sociology and Social Work, Vilnius University, Vilnius, Lithuania, 22 Department of Sociology, Charles University, Prague, Czech Republic, 23 School of Criminology, University of Los Andes, Mérida, Venezuela, 24 School of Social Sciences, University of Iceland, Reykjavik, Iceland, 25 Department of Social Work, Criminology and Public Health Sciences, University of Gävle, Gävle, Sweden, 26 School of Criminology and Criminal Justice, Northeastern University, Boston, Massachusetts, United States of America

* david.builgil@manchester.ac.uk

## Abstract

This study investigates whether unstructured spare time is a significant correlate of self-reported offending among adolescents in multiple countries. Drawing on survey data from 58,425 13-to-17-year-olds in 21 countries in Europe, North America, and South America, we examine whether time spent in unstructured out-of-home and at-home activities, as well as structured time at home, is associated with offending prevalence and incidence. Using multivariate models that control for key criminological predictors, we find that unstructured out-of-home spare time is a robust and consistent correlate of self-reported offending. Its estimated association is larger than that of most classical predictors. While structured spare time at home is associated

**Data availability statement:** Replication data for this study are available via Zenodo (https://doi.org/10.5281/zenodo.19596718). This Zenodo record provides access upon request to the dataset used in this study for the 21 countries analyzed. Access is currently provided on a restricted basis in accordance with the terms agreed with study participants and relevant ethical committees until the full International Self-Report Delinquency Study 4 (ISRD4) dataset becomes openly available. The full raw data from ISRD4 remain subject to an embargo agreed with study participants and relevant ethical committees. Following the terms agreed with study participants and specified in ethical approvals across participating countries, the complete ISRD4 dataset will be made openly available without restriction at the end of 2026 or beginning of 2027. All analytic code and relevant data documentation are available on GitHub (https://github.com/davidbuilgil/unstructured-spare-time-crime).

**Funding:** In Argentina, this study was funded by the Scientific and Technological Research Fund (PICT 2019 call) and by the Office of Science and Technology at the National University of Córdoba (PIDTA 2023 call). Data collection in the UK was funded by the Nuffield Foundation, grant number JUS/FR-000022640. The Bosnia and Herzegovina research team extends special thanks to the Zurich University of Applied Sciences for supporting data collection through the Swiss National Science Foundation (Project No. 192539). The Slovenian part of the research was supported by the Slovenian Research and Innovation Agency (ARIS), project on Local Safety and Security – comparison between rural and urban environments, and its subsection on self-reported juvenile delinquency and victimization (Grant No. P5-0397). Patrik Manzoni and Sandrine Haymoz thank the Swiss National Science Foundation (Project No. 192539) for funding the survey conducted in Switzerland and the Balkan countries. The ISRD4 study in Estonia was supported by the project 'Establishing a Special Treatment System for Juveniles,' funded under the program 'Local Development and Poverty Reduction' of the European Economic Area and Norway Grants 2014–2021 (Grant Agreement No. 7-8/3771), administered by the Estonian Ministry of Justice and implemented by the University of Tartu. The ISRD4

with lower levels of offending, unstructured spare time, particularly out-of-home, is strongly linked to both prevalence and incidence of crime involvement. Country-specific analyses reveal that this pattern holds across most national samples. Simulation analyses suggest that modest reductions in unstructured out-of-home spare time may be associated with lower levels of adolescent offending. These findings indicate that unstructured time environments constitute a cross-culturally robust correlate of adolescent offending, with potential relevance across dispositional and opportunity-based explanations, and with implications for how prevention frameworks conceptualize adolescents' everyday environments.

## Introduction

Crime remains a major challenge for contemporary societies, affecting approximately 12% of the world population each year through property crime and 6% through violence [1]. Globally, 52 homicides are recorded each hour, resulting in five times more deaths than armed conflicts and twenty times more deaths than terrorism [2]. A significant share of this violence involves younger people: approximately 44% of individuals brought into formal contact for homicide are aged 15–29, despite this age group comprising only 23% of the global population [2]. The consequences of crime extend beyond individual victims, shaping economic development, societal inequalities, and psychosocial wellbeing [3–5].

To understand and prevent crime, scholars have developed a wide range of theories that emphasize different aspects of criminal behavior. Classic perspectives focus on social learning [6], social and self-control [7,8], moral development [9], opportunity structures [10, 11], and the role of community disorganization [12,13]. These theoretical frameworks have helped explain why young people become involved in crime, how offending patterns evolve over the life-course [14–16], and why crime concentrates in specific contexts [12,17]. Yet, most of these theories have paid relatively limited explicit attention to how adolescents' everyday time use structures their exposure to criminogenic or protective environments. Although situational perspectives emphasize the importance of "where, when, and with whom" activities take place, time use itself has often been treated as a background condition rather than as a central explanatory factor. A growing body of research has begun to examine adolescents' leisure routines empirically [18–22], but this work has largely been confined to single-country studies conducted in high-income contexts. As a result, it remains unclear whether the association between unstructured time use and offending reflects context-specific institutional or cultural conditions, or whether it represents a more general pattern that holds across diverse social settings. Testing this relationship across countries is essential for assessing its cross-cultural robustness and its relevance for comparative theory and policy discussions.

This article presents international evidence that individual differences in adolescent crime are significantly associated with unstructured spare time, defined as time spent on activities for amusement, relaxation, or entertainment without predetermined

study in Brazil was partially supported by the São Paulo Research Foundation (FAPESP), under grants #2021/04732-2, #2022/10125-4, #2022/13907-3, and #2024/00906-4. There was no additional external funding received for this study. The funders had no role in study design, data collection and analysis, decision to publish, or preparation of the manuscript.

**Competing interests:** The authors declare no competing interests.

agendas or goals [23–26]. Examples include hanging out on the streets, attending parties, playing video games, and browsing social media, both at home and out-of-home. While spare time is known to offer numerous benefits for psychosocial development and wellbeing [27–29], under certain conditions—i.e., when it is *unstructured* and *extended in duration*—it has been linked to elevated risk of criminal involvement [18–21]. A recent U.S. study, for example, found that unstructured leisure time mediates part of the effects of parental supervision, social bonds, self-control, and peer influence on adolescent crime, emerging as a key explanatory factor [22]. Time spent in unstructured activities may reduce supervision, increase exposure to deviant peers, and create unsupervised opportunities for offending. In contrast, structured spare time—such as educational or family routines, sports, artistic or cultural group spaces, and religious participation—tends to foster prosocial values and promote long-term goal pursuit [30–35]. These dynamics suggest that the organization and context of adolescents' time use may play an important role in shaping behavioral outcomes. Further discussion of how unstructured spare time is conceptualized in this study, how it extends beyond Osgood et al.'s notion of "unstructured socializing," [24] and the mechanisms linking time use to offending is provided in Appendix S2 in S1 File.

Building on this conceptual and empirical foundation, the present study investigates whether variations in unstructured spare time are associated with self-reported offending across diverse national contexts. In doing so, the article assesses the extent to which unstructured spare time operates as a general cross-national correlate of adolescent offending, with potential relevance across dispositional and opportunity-based explanations. Using a sample of 58,425 adolescents aged 13–17 from 21 countries in Europe, North America, and South America [36], we ask: 'Is unstructured spare time a significant correlate of self-reported offending across multiple countries?' We distinguish between unstructured time spent at home and out-of-home, examine both offending prevalence and incidence, and assess the robustness of these relationships after adjusting for key criminological predictors. Comparative analysis is essential to assess whether this relationship holds consistently across different cultural, social, and institutional environments.

## Materials and methods

The study uses a sample of 58,425 13-to-17-year-old students across 21 countries (i.e., Argentina, Austria, Bosnia and Herzegovina, Brazil, Colombia, Czech Republic, Denmark, Estonia, Finland, Iceland, Lithuania, Mexico, Norway, Poland, Slovenia, Spain, Sweden, Switzerland, United Kingdom, United States, and Venezuela), with an average sample size of 2,782 students per country (min = 1,137 in Denmark; max = 11,880 in Switzerland). Multi-stage sampling was followed to select two or more cities (or regions) in each country, schools within cities, and classes within schools. From the sampled schools, entire classes were selected, and all students who were present on the day of the survey and who provided assent (alongside appropriate parental consent, where required) were invited to participate. Given that samples were drawn from urban areas only, cross-national comparisons reflect differences

between sampled cities rather than entire countries. Questionnaires were administered via computers in classroom settings, except in Venezuela, Argentina, and certain areas of Bosnia and Herzegovina and Slovenia, where they were administered on paper due to insufficient infrastructure for computer-assisted surveys. Data were collected between 1 January 2021 and 31 May 2024, with timelines varying slightly between countries. The research team accessed the data for research purposes on 17 February 2025, with all participant-identifying information removed. Data collection was part of the International Self-Report Delinquency Study 4 (ISRD4) [36]. Further details about the sampling design and data collection are available in Appendix S1 in S1 File.

In order to address our core research question, we model the association of *unstructured* out-of-home spare time, *unstructured* spare time at home, and *structured* spare time at home, as opposed to non-spare time, with self-reported offending. Ordinal frequency categories for time-use responses were translated into weekly hours using fixed multipliers (e.g., 'once a week', 'every day'), aggregated into time-use indices, and standardized to 24 hours after accounting for all activities in which respondents were engaged each day/week. *Unstructured out-of-home spare time* measures hours spent hanging about the streets and in shopping centers, attending parties in the evening, and skipping class (Min = 0.0; Mean = 0.7; Median = 0.5; Max = 5.5; SD = 0.7). *Unstructured spare time at home* measures hours spent playing online games, browsing through social media, visiting sites for adults, navigating the darknet, and gambling online (Min = 0.0; Mean = 6.0; Median = 7.2; Max = 18.2; SD = 3.8). *Structured spare time at home* measures hours spent looking for information online for school or work, studying or doing homework, and having meals with family (Min = 0.0; Mean = 3.6; Median = 3.3; Max = 11.7; SD = 2.2). *Non-spare time* refers to hours in school, working, or sleeping (Min = 4.4; Mean = 13.7; Median = 13.0; Max = 24.0; SD = 3.5). The survey did not include measures of structured out-of-home spare time.

*Self-reported offending* considers all self-reported instances of damage (vandalism and graffiti), property crime (shoplifting, burglary, and vehicle theft), violence (robbery, group fights, and assault), drug dealing, and cybercrime (publicizing intimate images of someone without their consent, cyber hate, cyber fraud, and hacking) in the last 12 months. We first analyze offending prevalence (i.e., at least one incident) and then incidence (i.e., the number of incidents). To mitigate the influence of outliers in the sample, offence counts were calculated after removing values identified as outliers for each crime type—defined as those with an expected frequency of less than 0.5 under a fitted negative binomial distribution (915 respondents; 1.57% of the sample). Appendix S4 in S1 File provides detailed descriptive statistics for our dependent, independent, and control variables.

Beyond analyzing the association between time use on self-reported offending, our models control for a number of factors, including five key criminological variables: self-control (derived from four items combined via Confirmatory Factor Analysis (CFA)), parental control (four items combined via CFA), morality (six items combined via CFA), having delinquent peers (binary) and exposure to crime (five items combined via CFA); and five controls (gender (male), age, born in country, perceived family deprivation, and openness to be sincere in responding). Openness to respond sincerely is measured using the item "imagine you had shoplifted; do you think that you would have said so in this survey?" [36,37]. Appendix S3 in S1 File displays the four CFA models with the items considered and detailed factor estimates and covariances. Countries are included in our models as fixed effects. Appendix S5 in S1 File displays the country-level means of the key independent and dependent variables.

Two sets of models are estimated, first binary logit models to estimate the prevalence of self-reported offending, and second quasi-Poisson models to estimate the over-dispersed outcome of self-reported crime counts. Our binary logit model follows this formula:

$$\log\left(\frac{P(SRO \geq 1)}{P(SRO = 0)}\right) = \beta_0 + \sum_{i=1}^{13} \beta_i \times X_i + \sum_{k=1}^{K-1} \gamma_k \times I\left(c_k\right),$$

where $\log\left(\frac{P(SRO \geq 1)}{P(SRO=0)}\right)$ is the log-odds (logit) of the probability of self-reported offending being 1 or more as opposed to 0, $\beta_o$ is the intercept term, $\beta_1, \beta_2, \ldots, \beta_{13}$ are the coefficients for the key independent and control variables of the study, and $\sum_{k=1}^{K-1} \gamma_k \times I(c_k)$ represents the dummy variable for the categorical variable of country, $c_k$, with $K$ levels, where $I(c_k)$ indicates 1 if the unit belongs to the $k$-th country and 0 otherwise. The sum is over $K-1$ to avoid multicollinearity with the intercept.

Our quasi-Poisson model is given by:

$$\log(SRO) = \beta_0 + \sum_{i=1}^{13} \beta_i \times X_i + \sum_{k=1}^{K-1} \gamma_k \times I(c_k),$$

where $\log(SRO)$ is the natural logarithm of the expected count of self-reported offending. We additionally estimated negative binomial regression models to assess the sensitivity of the results to alternative count model specifications and present these as robustness checks in Appendix S7 in S1 File.

To account for the nested structure of the data (students within schools), standard errors are clustered at the school level, and all p-values are based on robust sandwich estimators [38]. Country fixed effects are included to adjust for unobserved between-country differences [39]. We estimate the models for all respondents first, and second for each country independently. This staged approach enables us to examine national associations directly and compare them across countries without constraining country-level estimates to follow a common distribution.

Effect sizes are presented as odds ratios or incidence rate ratios based on models in which quasi-continuous predictors were standardized (z-scores divided by two [40]), allowing for a more meaningful comparison of coefficients across variables originally measured on different scales.

The analysis has been programmed in R Software, and all analytic code is available on Github (https://github.com/davidbuilgil/unstructured-spare-time-crime). Replication data for the study are available via Zenodo (https://doi.org/10.5281/zenodo.19596718).

## Ethics statement

Each national team complied with relevant national legislation and ethical regulations for data collection. Informed consent was obtained from all participants prior to survey administration in accordance with the Declaration of Helsinki. Ethical approval for data collection was granted by the relevant bodies in participating countries, including: University of Iceland Ethics Committee for Scientific Research (SHV2022−034); National Council of Scientific and Technological Research and University of Córdoba (21/08/2020, 12/04/2022, 21/12/2022); University of Helsinki Ethical Review Board (43/2021); Council for Scientific, Humanistic, Technological and Arts Development of the University of los Andes (D-495-23-09-B); University of Salford Research Ethics Panel (2164); University of São Paulo Research Ethics Committee (51278521.6.0000.5407) and National Commission for Research Ethics (5.855.133); University of Maribor Ethics Committee (17/05/2022); Ministry of Education of Canton Sarajevo (11-03-02-34-46543/21) and Ministry of Education, Science, Culture and Sport of Una-Sana Canton (10-34-1286-4/22); Research Ethics Committee at Charles University (UKFF/13765/2023); Social Research Ethics Committee of University of Castilla-La Mancha (CEIS-638819-P8D2); Zurich University of Applied Sciences Rector's Office Legal Department (100017_192539); Institutional Review Board at University of New Haven (2023−055); Research Ethics Committee at University of Guadalajara (01/2022); and Research Ethics Committee at University of Medellín (report #48). Formal exemptions from ethics approval were granted by the Regional Ethical Review Board in Uppsala (2013/425) and the Norwegian Research Information Repository (929610). In Austria, Lithuania, Poland, and Estonia, formal ethics approval was not required for anonymous, non-interventional school-based social science research under applicable national or institutional regulations. In some cases, approvals or exemptions covered data collection across multiple countries.

## Results

We begin by descriptively examining whether participants with self-reported delinquent behavior have, on average, more unstructured spare time than those not involved in crime. As a first step, we consider unstructured spare time activities both at home and out-of-home. We compare the unstructured spare time of those with 0 counts of self-reported offending against 'low frequency' offenders (under or equal to the 95th percentile of self-reported offences in each national sample) and 'high frequency' offenders (over the 95th percentile) [41]. This comparison is essential to account for the fact that 'high frequency offenders' concentrate most crime incidents [16,42,43]. Welch-type pairwise t-tests are used to assess whether the differences displayed in Fig 1 are statistically significant, with p-values adjusted for multiple testing using the Benjamini-Hochberg procedure. In total, 63 pairwise comparisons were conducted (three within each national sample), with the Benjamini-Hochberg correction applied separately within each country.

High-frequency offenders report more unstructured spare time than non-offenders in every country, with the difference being statistically significant at the $p < 0.05$ level in all cases. Low frequency offenders also concentrate higher average scores of unstructured spare time than non-offenders in all countries, though such a difference is not statistically significant in Sweden. Finally, high frequency offenders report more unstructured spare time than low frequency offenders, but differences are only statistically significant in two-thirds of the countries (i.e., Austria, Bosnia and Herzegovina, Colombia, Czech Republic, Estonia, Finland, Iceland, Mexico, Poland, Slovenia, Spain, Switzerland, UK, and Venezuela). Overall, while non-offenders report spending, on average, 6.4 hours daily in unstructured spare time activities, the average is 7.5 for low frequency offenders and 8.8 for high frequency offenders. Results remain remarkably consistent when the 'low frequency' and 'high frequency' groups are separated using the 95th percentile of self-reported offences across the pooled national samples (i.e., 7 offences), as shown in Appendix S6 in S1 File. Further details on the country-level estimates of self-reported offending and unstructured spare time are available in Appendix S5 in S1 File.

Next, we examine whether variations in unstructured spare time, out-of-home and at home, and structured spare time at home are associated with individual differences in self-reported offending, measured as a binary (Model 1 in Table 1) and count outcome (Model 2 in Table 1), across all countries in the sample. The logit model estimates the binary outcome of offending prevalence, and the quasi-Poisson model estimates offending incidence. Our models control for other key variables, including self-control, parental control, morality, exposure to crime, and having delinquent peers, as well as demographic and national differences.

All three measures of individual time use are significantly associated with self-reported offending in both models ($p < 0.001$). Unstructured spare time, whether spent out-of-home or at home, is associated with increased self-reported offending. In contrast, structured spare time at home is associated with both lower prevalence and incidence of offending. Notably, the estimated association of unstructured out-of-home spare time is substantially larger than that of unstructured spare time at home. Across our two models, the only criminological control variable with a stronger association magnitude than unstructured out-of-home spare time is the binary measure of having delinquent peers; a well-established finding in the study of adolescent crime [6,19,26]. The estimated association of self-control is similar in size to that of unstructured out-of-home spare time; however, it operates in the opposite direction, being negatively associated with individual crime involvement. Parental control and morality are also negatively associated with self-reported offending, and exposure to crime has a positive association. Our models also control for openness in honestly reporting crime experiences in the survey, which is strongly correlated with offending outcomes. Including this variable helps reduce the risk that other observed effects are influenced by individual variation in social desirability [37]. The Negative Binomial models, whose results can be consulted in Appendix S7 in S1 File, produced substantively equivalent results to the main analyses.

Additional analyses presented in Appendix S8 in S1 File provide evidence that most of these relationships hold when predicting specific crime types (i.e., damage, property crime, violence, drug crime, and cybercrime) rather than the overarching measure of criminal offending. However, the association of unstructured out-of-home spare time is notably weaker

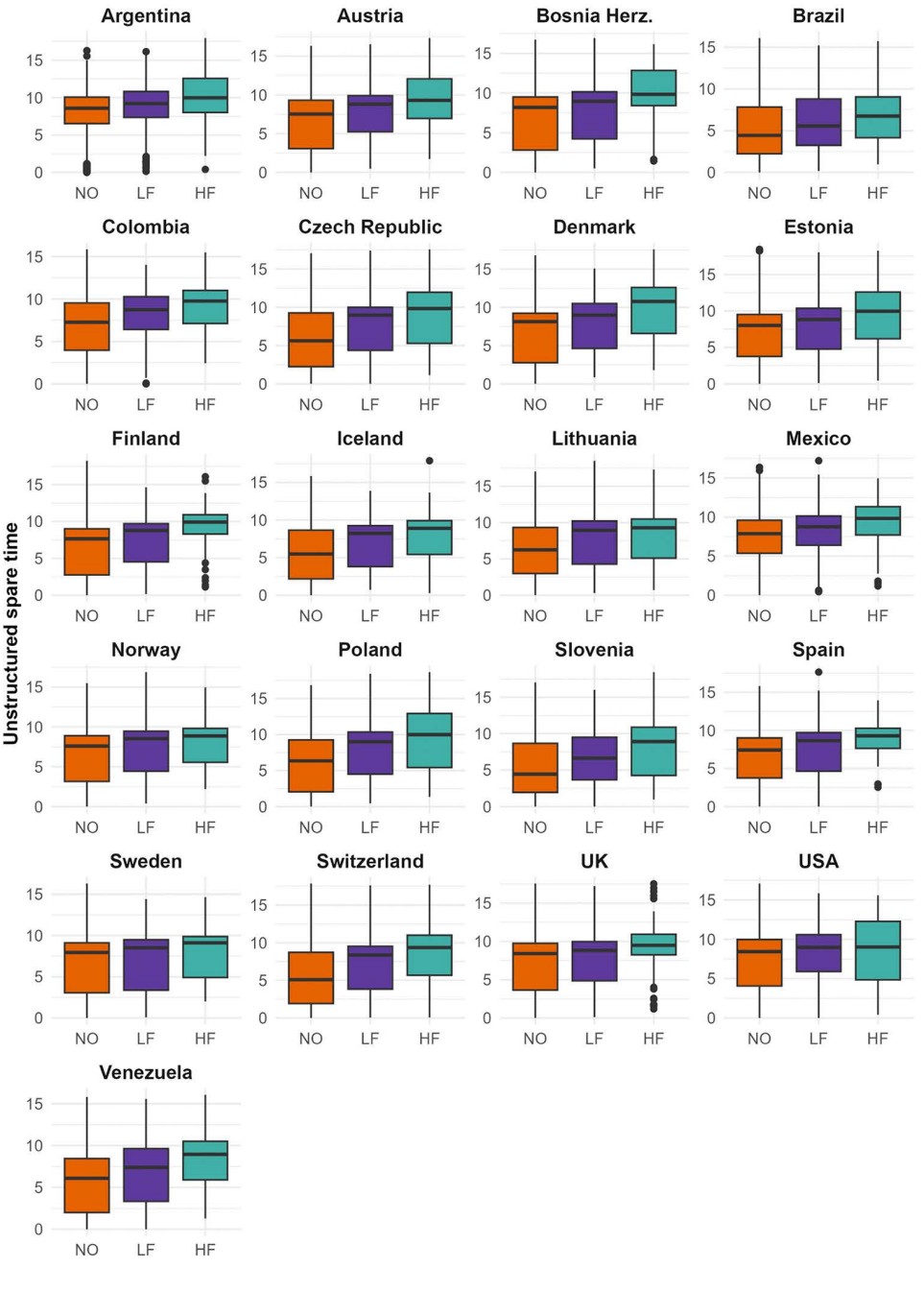

Fig 1. Boxplots of unstructured spare time (out-of-home and at home) across respondents with zero, low frequency and high frequency self-reported offending.

**Table 1. Binary logit model and quasi-Poisson model of self-reported offending.**

| | Model 1 Binary logit model | | | Model 2 Quasi-Poisson model | | |
|---|---|---|---|---|---|---|
| | OR | 95% CI | p-value | IRR | 95% CI | p-value |
| (Intercept) | **0.15** | 0.13-0.17 | 0.026 | **0.40** | 0.34-0.47 | <0.001 |
| Unstructured out-of-home spare time | **1.70** | 1.61-1.79 | <0.001 | **1.70** | 1.62-1.77 | <0.001 |
| Unstructured spare time at home | **1.33** | 1.24-1.41 | <0.001 | **1.29** | 1.21-1.37 | <0.001 |
| Structured spare time at home | **0.78** | 0.73-0.84 | <0.001 | **0.78** | 0.72-0.84 | <0.001 |
| Self-control | **0.57** | 0.54-0.60 | <0.001 | **0.61** | 0.58-0.64 | <0.001 |
| Parental control | **0.60** | 0.56-0.63 | <0.001 | **0.68** | 0.65-0.71 | <0.001 |
| Morality | **0.66** | 0.63-0.70 | <0.001 | **0.62** | 0.60-0.65 | <0.001 |
| Delinquent peers | **5.25** | 4.97-5.55 | <0.001 | **4.39** | 4.12-4.69 | <0.001 |
| Exposure to crime | **1.25** | 1.18-1.33 | <0.001 | **1.34** | 1.27-1.41 | <0.001 |
| Gender (male) | **1.39** | 1.32-1.47 | <0.001 | **1.63** | 1.54-1.72 | <0.001 |
| Age | **0.82** | 0.77-0.87 | <0.001 | **0.92** | 0.86-0.97 | 0.010 |
| Born in country | 1.03 | 0.94-1.13 | 0.511 | 0.94 | 0.87-1.02 | 0.230 |
| Family deprivation | **1.06** | 1.01-1.12 | 0.031 | 1.02 | 0.97-1.07 | 0.626 |
| Openness (sincerity) | **1.76** | 1.67-1.87 | <0.001 | **1.57** | 1.48-1.66 | <0.001 |
| Observations | 44,427 | | | 44,427 | | |
| $\chi^2$ test compared to null model | 10,224 (p<0.001) | | | 99,800 (p<0.001) | | |
| Pseudo $R^2$ SSE | 0.29 | | | 0.31 | | |
| Pseudo $R^2$ Nagelkerke | 0.39 | | | 0.90 | | |

Standardized coefficients. Country fixed effects are included but not displayed. Robust p-values are based on standard errors clustered at the school level. Pseudo $R^2$ values are provided for descriptive purposes but should not be interpreted as directly comparable across model families.

for cybercrime compared to offline offences. Further details and interpretations of these crime-specific analyses are provided in Appendix S8 in S1 File.

Having established a general association between unstructured spare time and self-reported offending in our sample, we turn our attention to exploring whether the relationship between unstructured spare time and crime is present across all countries in the sample. Fig 2 visually displays the standardized coefficient estimates of *unstructured* spare time out-of-home and at home, and *structured* spare time at home, in each national sample.

All countries except Lithuania demonstrate significant and strong positive associations between unstructured out-of-home spare time and either offending prevalence or incidence. In Colombia, unstructured out-of-home spare time is significantly associated with offending incidence but not prevalence; conversely, in Venezuela, it is significantly associated with prevalence but not incidence. All other countries in our sample show significant associations of unstructured out-of-home spare time across the two models. Countries with both high (e.g., Spain, Argentina) and moderate (e.g., Iceland, Norway) baseline offending levels display similarly strong estimated associations, suggesting that the predictive power of unstructured spare time is not confined to contexts with elevated rates of adolescent offending. The positive association of unstructured spare time at home is significant in ten countries for the binary measure of self-reported offending, and in eight countries for counts of offences. The negative association of structured spare time at home is statistically significant in seven countries for offending prevalence and in six countries for offending incidence. Overall, unstructured out-of-home spare time emerges as a more consistent and cross-nationally robust correlate of both offending prevalence and incidence than unstructured spare time at home or structured spare time at home.

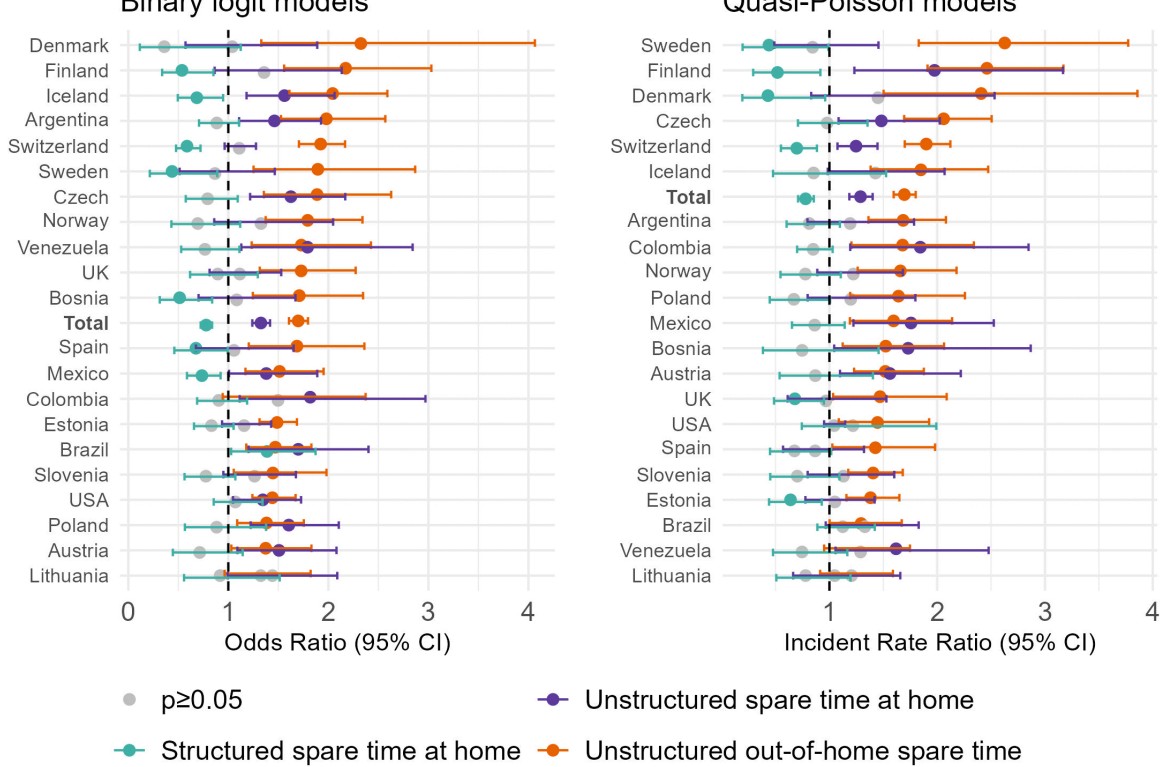

**Fig 2. Logistic and quasi-Poisson regression models of spare time and self-reported offending in each national sample.** Standardized coefficients. Estimates adjust for all key criminological and control variables. Robust p-values and confidence intervals are based on standard errors clustered at the school level.

For context, having delinquent peers is the only other variable in our national-level analyses that shows statistically significant associations with self-reported offending in all but one country. The association of delinquent peers with crime is consistently stronger than that of unstructured out-of-home spare time in every model. The association of morality is not statistically significant in Iceland, Lithuania, Mexico, Slovenia, and Sweden (prevalence), or in Denmark and Sweden (incidence), and is weaker than that of unstructured out-of-home spare time in all but five (prevalence) and eleven countries (incidence). Parental control is not significant in Austria (prevalence and incidence), Bosnia and Herzegovina (prevalence and incidence), Brazil (incidence), Denmark (incidence), Lithuania (prevalence and incidence), and Poland (incidence), and in many countries—seven in the logit and eight in the quasi-Poisson models—its estimated association is also weaker than that of unstructured out-of-home spare time. The association between self-control and self-reported offending is not statistically significant in four countries (prevalence) and eight countries (incidence), and among those with significant associations, it is weaker than that of unstructured out-of-home spare time in six (prevalence) and seven (incidence) countries. Lastly, exposure to crime is not significant in twelve (prevalence) and eleven countries (incidence). Overall, unstructured out-of-home spare time emerges as a stronger and more consistently correlate of self-reported offending than all other variables analyzed, with the sole exception of delinquent peer associations.

Estimates from our models can be utilized to generate model-based projections under hypothetical scenarios to simulate the impact of policies aimed at redirecting adolescents' unstructured spare time on overall self-reporting offending. We conducted bootstrapped simulations with 500 repeated samples to project the expected association between decreasing unstructured out-of-home spare time and offending prevalence and incidence, with results visually presented in Fig 3.

**Simulated Change in Offending Prevalence**

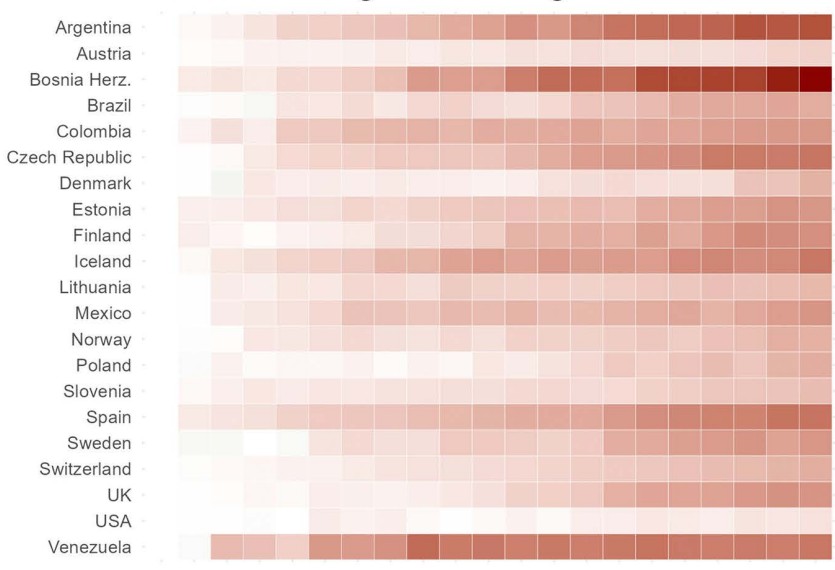

**Simulated Change in Offending Incidence**

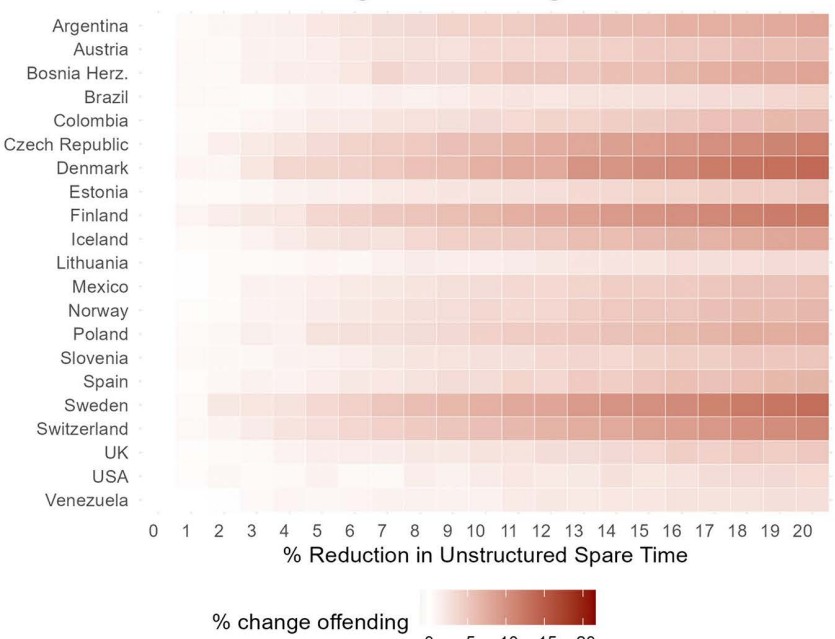

**Fig 3. Bootstrap simulations of projected changes in self-reported offending associated with reductions in unstructured out-of-home spare time.** 500 Bootstrap simulations.

Redirecting individuals' unstructured spare time by 10% is projected to be associated with an average reduction of 5.0% in offending prevalence (SD = 3.0) and 3.9% in offending incidence (SD = 1.8). A 20% reduction in unstructured out-of-home time is associated, on average, with a 9.6% decrease in prevalence (SD = 4.2) and a 7.4% decrease in incidence (SD = 3.3).

The projected reductions are particularly pronounced in countries such as Argentina, Bosnia and Herzegovina, Czech Republic, Denmark, Finland, Sweden, and Venezuela. For example, in Bosnia and Herzegovina, a 20% reduction in unstructured spare time is projected to be associated with a 22% lower prevalence, while Argentina would show a 16% decrease. Similarly, incidence is projected to lower by 14% in Denmark and 13% in Sweden upon a 20% reduction in unstructured spare time. These projected associations are not directly tied to baseline levels of offending; countries with both low (e.g., Sweden) and moderate (e.g., Bosnia and Herzegovina) rates of offending exhibit substantial projected declines.

## Discussion

This study provides robust cross-national evidence that unstructured spare time, especially when spent out-of-home, is strongly associated with adolescent self-reported offending. While unstructured leisure has received some attention in national-level research from high-income countries [e.g., 18–22], our findings extend this work by demonstrating that its association with offending is consistent across diverse cultural, economic, and social contexts, including those of low- and middle-income countries. Drawing on data from 21 countries, we find that unstructured activities are significantly associated with both the likelihood and frequency of offending, even after adjusting for established criminological factors such as parental control, self-control, morality, peer delinquency, and exposure to crime. In many countries, the association between unstructured out-of-home spare time and self-reported offending is comparable to or greater than those of these classical predictors; second only to the influence of delinquent peers. By contrast, structured spare time appears to offer a protective function, particularly when it takes place at home and involves activities that reinforce prosocial engagement [30–35]. Prior research indicates that unstructured spare time increases the risk of crime by weakening informal and formal social control, exposing adolescents to deviant peers, and creating unsupervised situations where opportunities for offending are more readily available [9,18,19,22,24,44,45]. Simultaneously, involvement in unstructured activities often means disengagement from structured, purposeful engagements that promote prosocial values and reinforce the pursuit of long-term goals. These findings suggest that the organization, content, and supervision of adolescents' leisure time play a critical role in shaping behavioral outcomes. The consistency of these associations across diverse national contexts further suggests that unstructured spare time captures a core feature of adolescents' routine activity environments that is not confined to specific institutional, cultural, or policy settings, but extends across markedly different social contexts.

While the cross-sectional nature of the data precludes causal claims and limits our ability to capture reverse causal mechanisms—such as the possibility that unstructured leisure options attract high-risk adolescents already involved in offending [46]—the consistency of findings across diverse national contexts, combined with the robustness of results after adjusting for key criminological controls, provides substantial evidence that unstructured spare time is an important correlate of adolescent crime and a potentially relevant risk marker for future longitudinal research. The cross-sectional nature of the data also limits our ability to distinguish between different developmental pathways, such as life-course persistent versus adolescence-limited offending [47], which remains an important area for future research.

Importantly, although our analyses draw on cross-nationally harmonized survey instruments, cross-national comparisons should be interpreted with appropriate caution. Differences in cultural norms, translation, and survey interpretation may affect the comparability of responses across national contexts. As such, some observed cross-national variation may reflect measurement differences in addition to substantive behavioral differences. Further research should also examine how broader socioeconomic inequalities shape patterns of unstructured spare time and offending across adolescents, communities, and countries. Moreover, it is important to note that our sample includes only students who are enrolled in school, excluding adolescents who may have even greater levels of unstructured spare time, suggesting that our findings may represent a conservative estimate of the relationship between unstructured time and offending.

It may be valuable for future studies to examine whether different forms of unstructured spare time, as well as structured out-of-home spare time (which could not be directly assessed in this study), differ in their associations with

offending. Our measures of time use combine numerous indicators into overarching groups of structured and unstructured behaviors, which may potentially obscure internal heterogeneity across specific activities [48,49]. Browsing social media, for example, may have less detrimental effects than navigating the darknet, and the level of formal control may vary between activities such as hanging about in the streets and attending parties. While assessing the relationship between each specific activity and offending is beyond the scope of this article, we highlight this as an important area for future research with potentially significant implications for crime prevention practice. To ensure that our results were not affected by the inclusion of spare time activities that may themselves be considered deviant rather than merely unstructured (i.e., truancy and navigating the darknet), we re-estimated the main analyses after excluding these items from the unstructured spare time measures. The results, shown in Appendix S9 in S1 File, remain substantively equivalent. Relatedly, the available data do not capture structured out-of-home activities such as sports participation, clubs, or organized after-school programs. Prior research has shown that engagement in structured out-of-home leisure activities may be linked to lower levels of delinquency [30,34,48–51]. Moreover, prior research has also reported that adolescents who are more involved in structured activities—particularly sports—may exhibit weaker associations between unstructured activities and delinquency [49]. Future research should examine how structured out-of-home activities compare with the forms of spare time analyzed here in their association with adolescent offending.

Theoretical and policy implications follow from these results. Theoretically, our findings highlight the need to better integrate dispositional and opportunity-based perspectives in the study of crime and youth delinquency [9]. The consistency of the relationship between unstructured spare time and offending across 21 countries with markedly different cultural, institutional, and socioeconomic contexts moves what has previously been a theoretical proposition into the status of a cross-cultural empirical regularity: one that no single-country study could have established, and one that therefore places a new demand on all major theoretical traditions. Time use sits at the intersection of everyday routines, individual predispositions, social bonds, and access to supervision and opportunities, and therefore offers a conceptual bridge across theoretical traditions rather than merely serving as a background variable within any one of them [52]. Notably, the finding that unstructured out-of-home spare time rivals self-control as a correlate of offending in many national samples, and outperforms parental control and morality in most, suggests that dispositional accounts alone are insufficient: the structure and supervision of daily activities shape the conditions under which dispositional traits translate into behavior [9,19,20,22]. This underscores the importance of studying how individual predispositions interact with situational contexts over time, particularly in shaping exposure to criminogenic versus prosocial environments [53,54].

From a policy standpoint, our findings suggest that prevention strategies focused on the availability and quality of adolescents' structured time warrant consideration, not as evaluations of specific interventions but as a reframing of how time-use environments are conceptualized within prevention debates. Interventions that seek to redirect unstructured spare time, particularly in public, outdoor settings, and expand access to structured, supervised, and meaningful activities may represent promising avenues for prevention, although their effectiveness should be evaluated using stronger causal designs. These efforts can operate across multiple levels, including families, communities, schools, and local governance. Parents and caregivers can be supported to foster structured routines at home, while communities can expand access to structured leisure opportunities through youth centers, sports, and arts programs [51]. Schools play a critical role through after-school initiatives and mentoring schemes [55–57], while exclusions limit access to structured environments and exacerbate risk [58]. Local authorities can improve access to safe public spaces and transport. Such initiatives should not aim to curtail adolescents' autonomy in out-of-home settings, but rather seek to expand the availability of structured, safe, and developmentally enriching options. Our findings are consistent with concerns that disinvestment in youth services may reduce access to structured leisure opportunities relevant to adolescent offending [59]. Simulation analyses suggest that even moderate changes in time use may be associated with meaningful reductions in offending. Digital environments also warrant attention, as unsupervised online time at home may present similar risks. Overall, these findings underscore

that how adolescents spend their time is strongly associated with youth delinquency and may represent a promising area for prevention-oriented intervention and future longitudinal research.

## Supporting information

**S1 File. Unstructured spare time as an international predictor of adolescent crime: Appendices S1-S9.** (DOCX)

## Acknowledgments

The authors would like to thank the ISRD4 national teams, school authorities, and pupils for making data collection possible. We are especially grateful to Günter Stummvoll for leading the data collection in Austria.

## Author contributions

**Conceptualization:** David Buil-Gil, Christopher Birkbeck.

**Data curation:** Christopher Birkbeck, Dirk Enzmann, Ineke Haen Marshall.

**Funding acquisition:** Christopher Birkbeck, Dirk Enzmann, Karin Arbach, Marina Rezende Bazon, Muhamed Budimlić, Danielle T. Cooper, Marta Dąbrowska, Esther Fernández-Molina, Carolyn Gentle-Genitty, Aurea E. Grijalva-Eternod, Ewa M. Guzik-Makaruk, Sandrine Haymoz, Neal Hazel, Markus Kaakinen, Janne Kivivuori, Iza Kokoravec Povh, Camilla Løvschall Langeland, Patrik Manzoni, Anna Markina, Gorazd Meško, Sandra Kobajica Mišanović, Kim Moeller, Solbey Morillo-Puente, Aušra Pocienė, Zuzana Podaná, Juan Antonio Rodríguez, Margret Valdimarsdottir, Lars Westfelt, Ineke Haen Marshall.

**Investigation:** David Buil-Gil, Christopher Birkbeck, Dirk Enzmann, Karin Arbach, Marina Rezende Bazon, Muhamed Budimlić, Danielle T. Cooper, Marta Dąbrowska, Esther Fernández-Molina, Carolyn Gentle-Genitty, Aurea E. Grijalva-Eternod, Ewa M. Guzik-Makaruk, Sandrine Haymoz, Neal Hazel, Markus Kaakinen, Janne Kivivuori, Iza Kokoravec Povh, Camilla Løvschall Langeland, Patrik Manzoni, Anna Markina, Gorazd Meško, Sandra Kobajica Mišanović, Kim Moeller, Solbey Morillo-Puente, Aušra Pocienė, Zuzana Podaná, Juan Antonio Rodríguez, Margret Valdimarsdottir, Lars Westfelt, Ineke Haen Marshall.

**Methodology:** David Buil-Gil.

**Project administration:** Christopher Birkbeck, Dirk Enzmann, Ineke Haen Marshall.

**Software:** David Buil-Gil.

**Visualization:** David Buil-Gil.

**Writing – original draft:** David Buil-Gil, Christopher Birkbeck.

**Writing – review & editing:** Dirk Enzmann, Karin Arbach, Marina Rezende Bazon, Muhamed Budimlić, Danielle T. Cooper, Marta Dąbrowska, Esther Fernández-Molina, Carolyn Gentle-Genitty, Aurea E. Grijalva-Eternod, Ewa M. Guzik-Makaruk, Sandrine Haymoz, Neal Hazel, Markus Kaakinen, Janne Kivivuori, Iza Kokoravec Povh, Camilla Løvschall Langeland, Patrik Manzoni, Anna Markina, Gorazd Meško, Sandra Kobajica Mišanović, Kim Moeller, Solbey Morillo-Puente, Aušra Pocienė, Zuzana Podaná, Juan Antonio Rodríguez, Margret Valdimarsdottir, Lars Westfelt, Ineke Haen Marshall.

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
