## [Decision Letter · Decision Letter 0]

12 Apr 2026

PONE-D-26-07062Unstructured Spare Time as an International Predictor of Adolescent CrimePLOS One

Dear Dr. Buil-Gil,

Thank you for submitting your manuscript to PLOS ONE. The reviewers recommend minor revisions. Therefore, we invite you to submit a revised version of the manuscript that addresses the points raised during the review process.

This manuscript presents a strong and valuable contribution, drawing on an exceptionally large cross-national dataset to examine the relationship between unstructured spare time and adolescent offending. The analyses are generally well-executed, and the consistency of findings is particularly compelling.

However, a small number of issues should be addressed before publication. In particular, the authors should (1) adopt more cautious, non-causal language when interpreting results, (2) clarify and justify the operationalisation of unstructured spare time, (3) improve transparency around methodological choices (including model specification and statistical tests), and (4) briefly discuss limitations related to cross-national comparability and sample representativeness. Minor structural improvements (e.g., positioning of the methods section) would also enhance clarity of this already good manuscript.

These are relatively minor revisions that will strengthen the manuscript without requiring substantial reanalysis. Subject to these, the article is suitable for publication.

We look forward to receiving your revised manuscript.

Kind regards,

Angelo Moretti, Ph.D.

Academic Editor

PLOS One

**Journal Requirements:**

1. When submitting your revision, we need you to address these additional requirements. Please ensure that your manuscript meets PLOS ONE's style requirements, including those for file naming. The PLOS ONE style templates can be found at https://journals.plos.org/plosone/s/file?id=wjVg/PLOSOne_formatting_sample_main_body.pdf and https://journals.plos.org/plosone/s/file?id=ba62/PLOSOne_formatting_sample_title_authors_affiliations.pdf 2. We note that the grant information you provided in the ‘Funding Information’ and ‘Financial Disclosure’ sections do not match.  When you resubmit, please ensure that you provide the correct grant numbers for the awards you received for your study in the ‘Funding Information’ section. 3. Thank you for stating in your Funding Statement: In Argentina, this study was funded by the Scientific and Technological Research Fund (PICT 2019 call) and by the Office of Science and Technology at the National University of Córdoba (PIDTA 2023 call). Data collection in the UK was funded by the Nuffield Foundation, grant number JUS/FR-000022640. The Bosnia and Herzegovina research team extends special thanks to the Zurich University of Applied Sciences for supporting data collection through the Swiss National Science Foundation (Project No. 192539). The Slovenian part of the research was supported by the Slovenian Research and Innovation Agency (ARIS), project on Local Safety and Security – comparison between rural and urban environments, and its subsection on self-reported juvenile delinquency and victimization (Grant No. P5-0397). Patrik Manzoni and Sandrine Haymoz thank the Swiss National Science Foundation (Project No. 192539) for funding the survey conducted in Switzerland and the Balkan countries. The ISRD4 study in Estonia was supported by the project ‘Establishing a Special Treatment System for Juveniles,’ funded under the program ‘Local Development and Poverty Reduction’ of the European Economic Area and Norway Grants 2014–2021 (Grant Agreement No. 7-8/3771), administered by the Estonian Ministry of Justice and implemented by the University of Tartu. The ISRD4 study in Brazil was partially supported by the São Paulo Research Foundation (FAPESP), under grants #2021/04732-2, #2022/10125-4, #2022/13907-3, and #2024/00906-4. Please provide an amended statement that declares *all* the funding or sources of support (whether external or internal to your organization) received during this study, as detailed online in our guide for authors at http://journals.plos.org/plosone/s/submit-now. Please also include the statement “There was no additional external funding received for this study.” in your updated Funding Statement. Please include your amended Funding Statement within your cover letter. We will change the online submission form on your behalf. 4. Thank you for stating the following financial disclosure: In Argentina, this study was funded by the Scientific and Technological Research Fund (PICT 2019 call) and by the Office of Science and Technology at the National University of Córdoba (PIDTA 2023 call). Data collection in the UK was funded by the Nuffield Foundation, grant number JUS/FR-000022640. The Bosnia and Herzegovina research team extends special thanks to the Zurich University of Applied Sciences for supporting data collection through the Swiss National Science Foundation (Project No. 192539). The Slovenian part of the research was supported by the Slovenian Research and Innovation Agency (ARIS), project on Local Safety and Security – comparison between rural and urban environments, and its subsection on self-reported juvenile delinquency and victimization (Grant No. P5-0397). Patrik Manzoni and Sandrine Haymoz thank the Swiss National Science Foundation (Project No. 192539) for funding the survey conducted in Switzerland and the Balkan countries. The ISRD4 study in Estonia was supported by the project ‘Establishing a Special Treatment System for Juveniles,’ funded under the program ‘Local Development and Poverty Reduction’ of the European Economic Area and Norway Grants 2014–2021 (Grant Agreement No. 7-8/3771), administered by the Estonian Ministry of Justice and implemented by the University of Tartu. The ISRD4 study in Brazil was partially supported by the São Paulo Research Foundation (FAPESP), under grants #2021/04732-2, #2022/10125-4, #2022/13907-3, and #2024/00906-4.   Please state what role the funders took in the study.  If the funders had no role, please state: "The funders had no role in study design, data collection and analysis, decision to publish, or preparation of the manuscript." If this statement is not correct you must amend it as needed. Please include this amended Role of Funder statement in your cover letter; we will change the online submission form on your behalf. 5. We note that you have indicated that there are restrictions to data sharing for this study. For studies involving human research participant data or other sensitive data, we encourage authors to share de-identified or anonymized data. However, when data cannot be publicly shared for ethical reasons, we allow authors to make their data sets available upon request. For information on unacceptable data access restrictions, please see http://journals.plos.org/plosone/s/data-availability#loc-unacceptable-data-access-restrictions.  Before we proceed with your manuscript, please address the following prompts: a) If there are ethical or legal restrictions on sharing a de-identified data set, please explain them in detail (e.g., data contain potentially identifying or sensitive patient information, data are owned by a third-party organization, etc.) and who has imposed them (e.g., a Research Ethics Committee or Institutional Review Board, etc.). Please also provide contact information for a data access committee, ethics committee, or other institutional body to which data requests may be sent. b) If there are no restrictions, please upload the minimal anonymized data set necessary to replicate your study findings to a stable, public repository and provide us with the relevant URLs, DOIs, or accession numbers. Please see http://www.bmj.com/content/340/bmj.c181.long for guidelines on how to de-identify and prepare clinical data for publication. For a list of recommended repositories, please see https://journals.plos.org/plosone/s/recommended-repositories. You also have the option of uploading the data as Supporting Information files, but we would recommend depositing data directly to a data repository if possible. Please update your Data Availability statement in the submission form accordingly. 6. Your ethics statement should only appear in the Methods section of your manuscript. If your ethics statement is written in any section besides the Methods, please move it to the Methods section and delete it from any other section. Please ensure that your ethics statement is included in your manuscript, as the ethics statement entered into the online submission form will not be published alongside your manuscript. 7. We note that there is identifying data in the Supporting Information file <clean_SI>. Due to the inclusion of these potentially identifying data, we have removed this file from your file inventory. Prior to sharing human research participant data, authors should consult with an ethics committee to ensure data are shared in accordance with participant consent and all applicable local laws. Data sharing should never compromise participant privacy. It is therefore not appropriate to publicly share personally identifiable data on human research participants. The following are examples of data that should not be shared: -Name, initials, physical address-Ages more specific than whole numbers-Internet protocol (IP) address-Specific dates (birth dates, death dates, examination dates, etc.)-Contact information such as phone number or email address-Location data-ID numbers that seem specific (long numbers, include initials, titled “Hospital ID”) rather than random (small numbers in numerical order) Data that are not directly identifying may also be inappropriate to share, as in combination they can become identifying. For example, data collected from a small group of participants, vulnerable populations, or private groups should not be shared if they involve indirect identifiers (such as sex, ethnicity, location, etc.) that may risk the identification of study participants. Additional guidance on preparing raw data for publication can be found in our Data Policy (https://journals.plos.org/plosone/s/data-availability#loc-human-research-participant-data-and-other-sensitive-data) and in the following article: http://www.bmj.com/content/340/bmj.c181.long. Please remove or anonymize all personal information (<specific identifying information in file to be removed>), ensure that the data shared are in accordance with participant consent, and re-upload a fully anonymized data set. Please note that spreadsheet columns with personal information must be removed and not hidden as all hidden columns will appear in the published file. 8. Please include captions for your Supporting Information files at the end of your manuscript, and update any in-text citations to match accordingly. Please see our Supporting Information guidelines for more information: http://journals.plos.org/plosone/s/supporting-information. 9. If the reviewer comments include a recommendation to cite specific previously published works, please review and evaluate these publications to determine whether they are relevant and should be cited. There is no requirement to cite these works unless the editor has indicated otherwise.

Reviewers' comments:

Reviewer's Responses to Questions

**Comments to the Author**

1. Is the manuscript technically sound, and do the data support the conclusions?

Reviewer #1: Yes

Reviewer #2: Yes

2. Has the statistical analysis been performed appropriately and rigorously? 

Reviewer #1: Yes

Reviewer #2: Yes

3. Have the authors made all data underlying the findings in their manuscript fully available?

Reviewer #1: Yes

Reviewer #2: No

4. Is the manuscript presented in an intelligible fashion and written in standard English?

Reviewer #1: Yes

Reviewer #2: Yes

5. Review Comments to the Author

**Reviewer #1:** This is an impressive and comprehensive study which draws on survey data from just under 60K young people across 21 countries. It explores the relationship between adolescent’s (aged between 13-17 years old) unstructured spare time and self-reported offending, for both the prevalence (probability of offending) and incidence (scale of that offending) of self-reported offending. The research shows that increased unstructured spare time out of the home is associated with higher levels of offending and is consistent across most countries in the study.

The paper addresses the relationship between unstructured spare time and adolescent offending with compelling results. The more unstructured spare time out of the home was associated with greater levels of offending, which makes sense. Especially, where there is also a lack of parental control and coupled with delinquent peers and greater exposure to crime. The model accounts for unstructured spare time at home and unstructured spare time out of the home, both of which increase the odds and scale of offending. The model also accounts for structured spare time at home, which has the opposite effect and reduces the odds and scale of offending. I therefore wondered, for completeness, what about the structured spare time out of the home? On lines 279-280 you say that the survey did not include measures of structured out of home spare time – that is such a shame. But, is there any way this could be approximated given the information we do know? If not, I think there is scope for some discussion to address this more explicitly. The focus of the paper is of course, the unstructured spare time and its effect on adolescent crime but given time is definitive,

The paper reports the results straight after the introduction, while the materials and methods section features at the end. The paper would benefit from moving the materials and methods section to before the results as with the normal convention of a research paper. This would make the study clearer from the outset and address the questions that arose as I read the results and discussion sections. The materials and methods section did answer all my questions that had but it would aid the interpretation of the results if it had featured earlier.

The study uses the International Self-Report Delinquency Study, a school-based survey instrument to measure self-reported offending. The supplementary material notes that the sample is not nationally representative and comparisons should be interpreted between cities rather than entire countries. For clarity, this could be made explicit within the main paper.

The study uses a Quasi-Poisson model to model the offending counts which is sensible. Out of curiosity, did the authors consider using a Negative-Binomial model? It would be interesting to assess the model fit and compare estimates from these two models.

**Reviewer #2:** The manuscript “Unstructured Spare Time as an International Predictor of Adolescent Crime” examines the relationship between adolescents’ time use and self-reported offending across 21 countries using a very large and valuable dataset. The paper addresses an important question in criminology and makes a potentially significant empirical contribution by moving beyond the usual single-country focus that dominates the literature on leisure and delinquency.

Overall, I find the paper interesting, well-executed in many respects, and clearly positioned within existing research, but I also believe that several conceptual and methodological issues need to be addressed before it can be considered for publication. In its current form, the manuscript tends to overinterpret its findings and would benefit from greater precision regarding what can (and cannot) be concluded from the data.

General assessment

One of the strongest aspects of the paper is the dataset itself. The use of the ISRD4 data, with more than 58,000 adolescents across a diverse set of countries, provides a level of external validity that is rarely achieved in this line of research. The consistency of the results across countries is also compelling and, in principle, supports the authors’ argument that unstructured spare time—particularly out-of-home—is a robust correlate of offending.

At the same time, the manuscript would benefit from a more cautious and theoretically grounded interpretation of these findings. The current framing sometimes suggests a level of causal inference that is not supported by the cross-sectional design, and some measurement and modeling choices require further clarification and justification.

On causal interpretation

The most important issue concerns the way the results are interpreted. Although the authors briefly acknowledge the cross-sectional nature of the data, several parts of the manuscript—particularly the Abstract, Discussion, and the simulation section—go beyond associational claims and imply causal effects. For example, the simulations suggesting that reducing unstructured spare time would lead to measurable reductions in offending rely on assumptions that cannot be justified with the current design.

This is problematic because alternative explanations are clearly plausible. Adolescents who are already involved in delinquency may be more likely to spend time in unstructured settings (selection effects), and unobserved factors such as family environment, neighborhood context, or personality traits could drive both time use and offending. While the inclusion of several control variables is a strength, it does not resolve these fundamental identification issues.

I would strongly encourage the authors to reframe the paper consistently in associational terms, reduce causal language throughout, and substantially qualify the interpretation of the simulation results. As it stands, the policy implications are too strong relative to the evidentiary basis.

On the measurement of unstructured spare time

A second major concern relates to how “unstructured spare time” is operationalized. The index combines a wide range of activities, some of which are conceptually quite different and may even overlap with deviant behavior itself. For example, including “skipping class” within the measure raises questions, as this is arguably already a form of rule-breaking rather than a neutral leisure activity. Similarly, the inclusion of heterogeneous online activities (from social media use to darknet browsing) blurs important distinctions.

This raises the possibility of conceptual and empirical overlap between the independent variable and the outcome, which could inflate the observed associations. At a minimum, the authors should provide a clearer justification for the inclusion of each component and discuss the implications of this measurement strategy.

On cross-national comparability

The cross-national scope is a major strength, but it also introduces important challenges that are not fully addressed. The analysis assumes that key constructs—such as unstructured time and self-reported offending—are comparable across countries. However, differences in cultural norms, reporting practices, and survey interpretation may affect both variables.

While the inclusion of country fixed effects helps control for average differences between countries, it does not address measurement equivalence. Without some discussion (or testing) of measurement invariance, it is difficult to assess whether the observed cross-national consistency reflects a substantive pattern or, at least in part, similarities in measurement artifacts.

I would therefore recommend that the authors discuss this issue of cross-national comparability, even if formal invariance testing is not feasible within the scope of the paper.

On modeling strategy

Relatedly, the modeling approach could be better justified. The data have a clear hierarchical structure (students nested within schools and countries), yet the analysis relies on clustered standard errors rather than multilevel models. While this is not necessarily incorrect, the choice should be explicitly justified, especially given the comparative focus of the study.

On additional methodological details

There are several smaller but still important points that would benefit from clarification:

• The use of pairwise t-tests with Benjamini–Hochberg correction is appropriate, but the implementation is not fully described (e.g., number of tests, whether correction is applied globally).

• The manuscript does not specify whether assumptions underlying the t-tests (particularly homogeneity of variances) were assessed, or whether Welch corrections were used.

• The variable capturing “openness in honestly reporting” is a potentially valuable addition, but its construction, validity, and relationship to other variables (such as morality or self-control) are not sufficiently explained.

These issues do not invalidate the results, but they reduce transparency and should be addressed.

In sum, this is a strong and potentially impactful paper that makes excellent use of a uniquely rich and large-scale cross-national dataset to address a highly relevant question in criminology. The empirical analyses are carefully conducted, and the consistency of the findings across countries is particularly compelling. The manuscript clearly demonstrates the promise of a time-use perspective for understanding adolescent offending and offers insights that are likely to be of broad interest to researchers and policymakers alike.

That said, the paper would benefit from some further refinement, particularly in clarifying aspects of measurement and modeling, adopting a slightly more cautious interpretation of the findings, and engaging more explicitly with issues of cross-national comparability. These are, in my view, constructive improvements that would strengthen an already valuable contribution.

6. PLOS authors have the option to publish the peer review history of their article (what does this mean?). If published, this will include your full peer review and any attached files.

Reviewer #1: No

Reviewer #2: **Yes:** Diego J. Maldonado-Guzmán

---

## [Author Response · Author response to Decision Letter 1]

27 Apr 2026

Dear Editor,

Thank you for your decision and for the opportunity to revise our manuscript. The reviewers’ feedback was insightful and helpful, and we have followed their recommendations in every instance.

As requested by the Managing Editor, we have also reworked the structure of the article, including moving the ethics statement to the Methods section, and formatted the manuscript in accordance with the journal’s guidelines.

We note that, in order to accommodate the revisions suggested by the reviewers, we have had to slightly shorten some sections of the article. We believe the revised manuscript is stronger as a result.

In the remainder of this memo, we outline the revisions made to the manuscript, listing each change alongside the relevant reviewer comment.

REVIEWER 1’S COMMENTS

Reviewer 1 commented that: “This is an impressive and comprehensive study which draws on survey data from just under 60K young people across 21 countries. It explores the relationship between adolescent’s (aged between 13-17 years old) unstructured spare time and self-reported offending, for both the prevalence (probability of offending) and incidence (scale of that offending) of self-reported offending. The research shows that increased unstructured spare time out of the home is associated with higher levels of offending and is consistent across most countries in the study.”

RESPONSE. We thank the Reviewer for their kind words about our study and appreciate their detailed and thoughtful comments. We respond below to each query raised and have done our best to revise the manuscript in line with the reviewer’s comments and suggestions.

Reviewer 1 observed that: “The paper addresses the relationship between unstructured spare time and adolescent offending with compelling results. The more unstructured spare time out of the home was associated with greater levels of offending, which makes sense. Especially, where there is also a lack of parental control and coupled with delinquent peers and greater exposure to crime. The model accounts for unstructured spare time at home and unstructured spare time out of the home, both of which increase the odds and scale of offending. The model also accounts for structured spare time at home, which has the opposite effect and reduces the odds and scale of offending. I therefore wondered, for completeness, what about the structured spare time out of the home? On lines 279-280 you say that the survey did not include measures of structured out of home spare time – that is such a shame. But, is there any way this could be approximated given the information we do know? If not, I think there is scope for some discussion to address this more explicitly. The focus of the paper is of course, the unstructured spare time and its effect on adolescent crime but given time is definitive.”

RESPONSE. This is an important consideration and we thank the reviewer for raising it. We agree that including structured out-of-home spare time activities in the analysis would substantially enhance the study’s implications for both theory development and, especially, crime prevention practise during adolescence. Activities such as sports programs or out-of-home after-school activities are not currently included in the analysis, even though they may strongly influence adolescents’ time use and opportunities to engage in crime. We thoroughly reviewed the survey items and found no measures that could reasonably approximate this information. We were unable to revise the analysis in this respect. We nevertheless agree that this is a priority for future research in the area and have therefore added the following explanation to the Discussion section:

“Relatedly, the available data do not capture structured out-of-home activities such as sports participation, clubs, or organized after-school programs. Prior research has shown that engagement in structured out-of-home leisure activities may be linked to lower levels of delinquency [30, 34, 48-51]. Moreover, prior research has also reported that adolescents who are more involved in structured activities—particularly sports—may exhibit weaker associations between unstructured activities and delinquency [49]. Future research should examine how structured out-of-home activities compare with the forms of spare time analyzed here in their association with adolescent offending.” (p. 17)

Reviewer 1 noted that: “The paper reports the results straight after the introduction, while the materials and methods section features at the end. The paper would benefit from moving the materials and methods section to before the results as with the normal convention of a research paper. This would make the study clearer from the outset and address the questions that arose as I read the results and discussion sections. The materials and methods section did answer all my questions that had but it would aid the interpretation of the results if it had featured earlier.”

RESPONSE. We appreciate the reviewer’s comment. Following this recommendation, the Methods section has now been moved immediately below the Introduction. All subsequent section and reference numbers have been modified accordingly.

Reviewer 1 asked: “The study uses the International Self-Report Delinquency Study, a school-based survey instrument to measure self-reported offending. The supplementary material notes that the sample is not nationally representative and comparisons should be interpreted between cities rather than entire countries. For clarity, this could be made explicit within the main paper.”

RESPONSE. This is an important consideration that was not sufficiently clear in the main manuscript. We have now added the following sentence to the Methods section to clarify this point:

“Given that samples were drawn from urban areas only, cross-national comparisons reflect differences between sampled cities rather than entire countries.” (p. 6)

Reviewer 1 commented that: “The study uses a Quasi-Poisson model to model the offending counts which is sensible. Out of curiosity, did the authors consider using a Negative-Binomial model? It would be interesting to assess the model fit and compare estimates from these two models.”

RESPONSE. We thank Reviewer 1 for raising this point. We had considered using a Negative Binomial regression to estimate crime counts, but selected a quasi-Poisson specification because it aligned better with the overdispersed nature of the data. Following the reviewer’s recommendation, however, we replicated the main analysis using a Negative Binomial model, and the results were substantively unchanged. We have added these robustness checks as Appendix S7 and included the following sentences in the main manuscript:

“We additionally estimated negative binomial regression models to assess the sensitivity of the results to alternative count model specifications and present these as robustness checks in Appendix S7.” (p. 8)

“The Negative Binomial models, whose results can be consulted in Appendix S7, produced substantively equivalent results to the main analyses.” (pp. 11-12)

REVIEWER 2’S COMMENTS

Reviewer 2 commented that: “The manuscript “Unstructured Spare Time as an International Predictor of Adolescent Crime” examines the relationship between adolescents’ time use and self-reported offending across 21 countries using a very large and valuable dataset. The paper addresses an important question in criminology and makes a potentially significant empirical contribution by moving beyond the usual single-country focus that dominates the literature on leisure and delinquency.

Overall, I find the paper interesting, well-executed in many respects, and clearly positioned within existing research, but I also believe that several conceptual and methodological issues need to be addressed before it can be considered for publication. In its current form, the manuscript tends to overinterpret its findings and would benefit from greater precision regarding what can (and cannot) be concluded from the data.

One of the strongest aspects of the paper is the dataset itself. The use of the ISRD4 data, with more than 58,000 adolescents across a diverse set of countries, provides a level of external validity that is rarely achieved in this line of research. The consistency of the results across countries is also compelling and, in principle, supports the authors’ argument that unstructured spare time—particularly out-of-home—is a robust correlate of offending.”

RESPONSE. We thank Reviewer 2 for their positive assessment of our study and for their detailed and thoughtful feedback. Below, we respond to each point raised and have revised the manuscript accordingly to reflect the reviewer’s comments and suggestions.

Reviewer 2 observed: “At the same time, the manuscript would benefit from a more cautious and theoretically grounded interpretation of these findings. The current framing sometimes suggests a level of causal inference that is not supported by the cross-sectional design, and some measurement and modeling choices require further clarification and justification.

The most important issue concerns the way the results are interpreted. Although the authors briefly acknowledge the cross-sectional nature of the data, several parts of the manuscript—particularly the Abstract, Discussion, and the simulation section—go beyond associational claims and imply causal effects. For example, the simulations suggesting that reducing unstructured spare time would lead to measurable reductions in offending rely on assumptions that cannot be justified with the current design.

This is problematic because alternative explanations are clearly plausible. Adolescents who are already involved in delinquency may be more likely to spend time in unstructured settings (selection effects), and unobserved factors such as family environment, neighborhood context, or personality traits could drive both time use and offending. While the inclusion of several control variables is a strength, it does not resolve these fundamental identification issues.

I would strongly encourage the authors to reframe the paper consistently in associational terms, reduce causal language throughout, and substantially qualify the interpretation of the simulation results. As it stands, the policy implications are too strong relative to the evidentiary basis.”

RESPONSE. We thank Reviewer 2 for raising this important point and agree with the concern expressed. We acknowledge that several parts of the original manuscript employed language that could be interpreted as implying a stronger level of causal inference than is warranted by the cross-sectional design. We have thoroughly revised the manuscript throughout to ensure that findings are framed consistently in associational rather than causal terms. We have replaced terms such as “effects,” “predictors,” and “risk factors” where appropriate, and clarify that the simulation analyses represent model-based projections rather than causal estimates of intervention impacts. We have also qualified the policy implications to ensure that they are presented more cautiously and proportionately to the design of the study. We provide below several examples of the revised sentences:

“This study investigates whether unstructured spare time is a significant correlate of self-reported offending among adolescents in multiple countries.” (p. 3)

“The estimated association of self-control is similar in size to that of unstructured out-of-home spare time; however, it operates in the opposite direction, being negatively associated with individual crime involvement.” (p. 11)

“All countries except Lithuania demonstrate significant and strong positive associations between unstructured out-of-home spare time and either offending prevalence or incidence.” (p. 13)

“For context, having delinquent peers is the only other variable in our national-level analyses that shows statistically significant associations with self-reported offending in all but one country.” (p. 14)

“Interventions that seek to redirect unstructured spare time, particularly in public, outdoor settings, and expand access to structured, supervised, and meaningful activities may represent promising avenues for prevention, although their effectiveness should be evaluated using stronger causal designs.” (p. 18)

In addition to these revisions, we have sharpened the conceptual framing in the Discussion's theoretical passage to make explicit that the paper's contribution is interpretive and theoretical rather than causal: the cross-national consistency of the association is presented as establishing a cross-cultural empirical regularity that places new demands on existing theoretical frameworks, rather than as causal evidence for intervention. This reinforces the associational stance emphasized throughout this response.

Reviewer 2 asked: “A second major concern relates to how “unstructured spare time” is operationalized. The index combines a wide range of activities, some of which are conceptually quite different and may even overlap with deviant behavior itself. For example, including “skipping class” within the measure raises questions, as this is arguably already a form of rule-breaking rather than a neutral leisure activity. Similarly, the inclusion of heterogeneous online activities (from social media use to darknet browsing) blurs important distinctions.

This raises the possibility of conceptual and empirical overlap between the independent variable and the outcome, which could inflate the observed associations. At a minimum, the authors should provide a clearer justification for the inclusion of each component and discuss the implications of this measurement strategy.”

RESPONSE. This is an important consideration and we appreciate the reviewer raising this point. In our view, here the reviewer’s comment raises two separate concerns: first, that our measures of unstructured spare time combine a range of different activities that may blur important distinctions both in their nature and in their relationship with crime; and second, that our measures of unstructured spare time include activities that may themselves be considered deviant and therefore may conceptually overlap with the dependent variables under study. We address these two concerns separately below.

Regarding the first point (i.e., that our measures of unstructured spare time combine a range of different activities that may mask important heterogeneity), we agree that this is an important consideration. It is indeed possible that our estimates average over activities with different implications for crime and crime prevention practice. However, the overall aim of the study was to assess whether unstructured spare time activities are, on average, more strongly related to self-reported offending across countries than structured activities. Analysing each activity separately would require a substantially expanded set of analyses and a different analytical focus from that of the present study. While we consider this beyond the scope of the present study, we acknowledge it as an important priority for future research and have added the following sentences to the Discussion section to clarify this point:

“Our measures of time use combine numerous indicators into overarching groups of structured and unstructured behaviors, which may potentially obscure internal heterogeneity across specific activities [48, 49]. Browsing social media, for example, may have less detrimental effects than navigating the darknet, and the level of formal control may vary between activities such as hanging about in the streets and attending parties. While assessing the relationship between each specific activity and offending is beyond the scope of this article, we highlight this as an important area for future research with potentially significant implications for crime prevention practice.” (p. 17)

Regarding the second point (i.e., that some items included in our unstructured spare time measures may themselves be considered deviant), we also agree that this is an important consideration and have re-run the analyses after excluding those measures of unstructured time use that may be considered deviant and thus c

---

## [Editor Report · Decision Letter 1]

28 Apr 2026

Unstructured spare time as an international predictor of adolescent crime

PONE-D-26-07062R1

Dear Dr. Buil-Gil,

We’re pleased to inform you that your manuscript has been judged scientifically suitable for publication and will be formally accepted for publication once it meets all outstanding technical requirements.

Kind regards,

Angelo Moretti, Ph.D.

Academic Editor

PLOS One
---

## [Editor Report · Acceptance letter]

PONE-D-26-07062R1

PLOS One

Dear Dr. Buil-Gil,

I'm pleased to inform you that your manuscript has been deemed suitable for publication in PLOS One. Congratulations! Your manuscript is now being handed over to our production team.

Kind regards,

on behalf of

Dr. Angelo Moretti

Academic Editor

PLOS One